# Stepping Further from Coupling Tools: Development of Functional Polymers via the Biginelli Reaction

**DOI:** 10.3390/molecules27227886

**Published:** 2022-11-15

**Authors:** Zeyu Ma, Bo Wang, Lei Tao

**Affiliations:** The Key Laboratory of Bioorganic Phosphorus Chemistry & Chemical Biology (Ministry of Education), Department of Chemistry, Tsinghua University, Beijing 100084, China

**Keywords:** the Biginelli reaction, functional polymer, bioactive polymer

## Abstract

Multicomponent reactions (MCRs) have been used to prepare polymers with appealing functions. The Biginelli reaction, one of the oldest and most famous MCRs, has sparked new scientific discoveries in polymer chemistry since 2013. Recent years have seen the Biginelli reaction stepping further from simple coupling tools; for example, the functions of the Biginelli product 3,4-dihydropyrimidin-2(1H)-(thi)ones (DHPM(T)) have been gradually exploited to develop new functional polymers. In this mini-review, we mainly summarize the recent progress of using the Biginelli reaction to identify polymers for biomedical applications. These polymers have been documented as antioxidants, anticancer agents, and bio-imaging probes. Moreover, we also provide a brief introduction to some emerging applications of the Biginelli reaction in materials and polymer science. Finally, we present our perspectives for the further development of the Biginelli reaction in polymer chemistry.

## 1. Introduction

Multicomponent reactions (MCRs) use three or more reactants to generate one single complex product in a one-pot manner. Thanks to pioneering studies, many MCRs have been used for polymer preparation. These MCRs includes the Passerini [1,2,3,4,5,6,7], Ugi [8,9,10,11,12,13,14,15,16,17], Biginelli [18,19,20,21,22,23,24,25,26], Hantzsch [27,28,29,30,31], and Kabachnik-Fields reactions [32,33,34,35,36], as well as alkyne-based MCRs [37,38,39] and metal-catalyzed MCRs [40,41,42,43]. Nowadays, MCRs have been widely acknowledged as handy tools for generating polymers with intriguing main-chain and side-chain structures.

Functions are the core of polymer research; synthesis strategies aiming to create new polymer structures may spark the appearance of new functional polymers. Recently, some polymers developed through MCRs have been applied in environmental science, materials science, and biomedical science [7,16,17,26,28,29,31,34,44,45,46,47,48,49,50]. In these studies, the use of MCRs to introduce various functional groups was demonstrated to be a major pathway to developing multi-functional polymers. Besides their use as efficient coupling tools, MCRs themselves generate functional structures, which open new opportunities to develop new functional polymers.

Among MCRs, the Biginelli reaction is one of the most famous, which was first reported by Italian chemist Pietro Biginelli in 1891 [51]. It is a tri-component reaction involving an aldehyde, a *β*-keto ester, and a (thio)urea to produce a 3,4-dihydropyrimidin-2(1H)-(thi)ones (DHPM(T)) heterocycle (Figure 1, down). The first step of the Biginelli reaction is the condensation between an aldehyde and a (thio)urea; then, the product reacts with a *β*-keto ester to furnish an additive product, which lastly cyclizes to form DHPM(T) under acid conditions. The Biginelli reaction offers robustness, easily accessible substrates, and mild conditions, with water as the only byproduct. Meanwhile, DHPM(T) derivatives have been documented as potential calcium antagonists, mitotic inhibitors, bacterial inhibitors, and antioxidants [52,53,54]. Therefore, the Biginelli reaction has been highly valued by chemists to effectively construct heterocycles and identify new drugs since its emergence.

The Biginelli reaction was introduced into polymer chemistry in 2009 [18], but it was not widely studied until 2013 [19]. In the early stage (2013–2016), researchers mainly focused on using the Biginelli reaction to develop polymer synthesis methodologies (Figure 1) [19,20,21,23,27,55]. Due to the deep understanding of this reaction in polymer chemistry, the bioactivities of DHPM(T) have been gradually exploited for developing new functional polymers; some unique properties and functions of Biginelli-type polymers have also been identified based on the synthesis methods (Figure 1) [21,22,25,26,56,57,58,59,60,61,62,63,64,65,66,67,68,69]. In this mini-review, we summarize the recent applications of the Biginelli reaction in developing functional polymers that can be used in the biomedical area as antioxidants, anticancer agents, and bioimaging probes. In addition, we also provide a brief introduction to other emerging applications of the Biginelli reaction in materials and polymer science. Finally, we present our perspectives on the future of the Biginelli reaction in polymer chemistry.

## 2. Bioactive Polymers

Oxidative stress occurs when excess reactive oxygen/nitrogen species (ROS/NOS) exist in the biological system [70]. These species attack biomacromolecules such as DNA, RNA, and proteins, thereby causing severe health issues. Oxidative stress has been recognized as being related to Alzheimer’s disease [71], cardiovascular disease [72], cancer [73], diabetes [74], and other diseases [75,76,77]. Antioxidants have been used to prevent or treat these diseases by regulating the uncontrolled redox system [78,79]. Compared to organic small molecule antioxidants that have limited clinical effects for the treatment of oxidative stress [80,81], polymeric antioxidants have enhanced in vivo stability, good water solubility, and increased bioavailability [82,83,84,85,86,87]. However, the direct introduction of small molecule antioxidants into polymers may cause a sharp loss of antioxidant capacity, resulting in an insufficient method for generating polymeric antioxidants. Thus, the development of new methodologies to efficiently produce polymeric antioxidants is important for both fundamental research and practical use.

DHPM(T) derivatives have been documented as radical scavengers since 2006 [88]. Moreover, owing to the modular nature of the Biginelli reaction, other antioxidant moieties can be facilely included to improve the antioxidation of the product. Therefore, the Biginelli reaction is a potential tool for exploring polymeric antioxidants. In 2017, Tao et al. developed a new strategy to obtain antioxidant polymers by introducing thiourea moieties and/or phenol moieties into polymers [25]. By combining ultra-fast reversible addition–fragmentation chain-transfer (RAFT) polymerization and the Biginelli reaction, they developed a high-throughput (HTP) strategy for preparing a library of 60 polymers containing different DHPM(T) moieties in the side chains (Figure 2a).

A 1,1-diphenyl-2-picrylhydrazyl (DPPH) radical-quenching experiment was conducted to test the radical-scavenging ability of the polymers. The ability of polymers to fade the purple color of DPPH reflected their antioxidant capacity (Figure 2b). The visual results and the quantitative data obtained by measuring the UV absorbance at 495 nm (Figure 2c) demonstrated that the polymers containing thiourea moieties (A(X)B(Y)C(2)) or/and phenol moieties (A(X)B(1)C(Z), A(X)B(3)C(Z) and A(X)B(4)C(Z)) efficiently quenched the DPPH radical. This study initially revealed the potential of the Biginelli reaction to develop antioxidant polymers.

Ultraviolet radiation (UVR) is a source of oxidative stress; radical scavengers may effectively control oxidative stress and protect biological systems from UVR damage [89]. Inspired by their previous work, in 2018, Tao et al. reported new antioxidant polymers developed by the Biginelli reaction that can protect cells from UVR [26]. They prepared 25 monomers using essential oil extracts, (thio)urea derivatives, and commercially available *β*-keto ester (2-acetoacetoxy)ethyl methacrylate (AEMA) in a mini-HTP manner; a library of 25 Biginelli-type polymers were facilely prepared through free radical polymerization (Figure 3a). The radical-scavenging ability of the polymers was evaluated; three monomers of polymers with the fastest DPPH-quenching rates were selected and copolymerized with poly(ethylene glycol) methyl ether methacrylate (PEGMA) to develop potential water-soluble UVR protectors. 

The as-prepared polymers were screened according to different criteria to obtain a single copolymer with low toxicity and that effectively protects cells from fatal UVR. The cytoprotective ability of the selected polymer was compared with superoxide dismutase (SOD) against UVR (Figure 3b). L929 cells cultured with the selected copolymer (10 mg/mL) or SOD (10 mg/mL) were exposed to a UV lamp (∼254 nm, 40 W). The Biginelli-type polymer excellently protected the cells after 15 min of UVR (300 ± 20 μW/cm^2^), while the SOD hardly protected the cells (Figure 3b). The mechanistic study revealed that the polymeric antioxidant may play an important role in preventing the DNA damage caused by UVR. These results demonstrate the utility of using the Biginelli reaction to develop polymeric UV protectors with practical use value. As the first facile preparation of a monomer/polymer library via the HTP-MCR strategy, this work also offers a new pathway for developing bioactive polymers.

To prompt the in-depth research of antioxidant polymers, in 2019, Tao et al. reported a ferrocene-containing antioxidant polymer prepared via the Biginelli reaction used to treat in vivo oxidative stress damage [58]. Ferrocene is a well-known reductant and free radical scavenger [90,91]. Owing to their low toxicity and versatile biological activities, ferrocene derivatives have been widely used as antimalarial and anticancer drugs [92,93]. The ferrocene group has been identified as capable of apparently enhancing the antioxidant ability of DHPMT derivatives [94]. In this study, Tao et al. used ferrocenecarboxaldehyde, thiourea, and AEMA as the substrates of the Biginelli reaction to obtain a DHPMT–ferrocene monomer, which was copolymerized with PEGMA to facilely produce a water-soluble polymer (P0, Figure 4a). This DHPMT–ferrocene polymer had better radical-scavenging ability than the polymers containing only ferrocene or DHPMT groups.

P0 was then evaluated in comparison to clinical drugs. Compared with small molecule antioxidants such as glutathione, P0 exhibited lower toxicity and offered much better protection to cells against tert-butyl hydroperoxide (*t*-BHP)-induced oxidative stress (Figure 4b). Moreover, in an in vivo experiment using mice as model animals, P0 was superior to silymarin, an active pharmaceutical ingredient in clinically prescribed medicines in the treatment of CCl_4_-induced acute liver damage (Figure 4c). Firstly, this research combines ferrocene and DHPMT groups into one polymer to achieve synergism for antioxidation, showing the value of the Biginelli reaction in developing new functional polymers for biomedical applications. Furthermore, this study may also inspire the application of organometal chemistry and MCRs to functional polymer synthesis.

The antioxidant property of DHPMT moiety has also been incorporated in multi-functional biomaterials. In 2020, Tao et al. developed an antioxidant self-healing hydrogel via the Biginelli reaction (Figure 5a) [63]. They prepared a monomer containing both phenylboronic acid (PBA) and DHPMT groups, which was copolymerized with PEGMA to produce a PBA–DHPMT polymer. The resulting polymer quickly cross-linked poly(vinyl alcohol) (PVA) to form a hydrogel under mild conditions (pH = 7.4, 25 °C). Since the borate ester bonds between PBA and the diol groups in PVA are dynamic, the resulting hydrogel was self-healable. Besides antioxidant and self-healing properties, this hydrogel showed low cytotoxicity and was successfully used as a 3D cell culture matrix (Figure 5b). Moreover, it demonstrated excellent bio-safety in a living mouse model. These results suggest that this newly-developed hydrogel may be a potential implant material and demonstrate the advantages of the Biginelli reaction in exploiting multi-functional biomaterials.

The PBA–DHPMT-containing polymer has also been exploited as a multi-functional bio-platform. Bacterial keratitis (BK) is a rapidly progressive and highly aggressive infectious corneal disease [95,96,97]. The inflammatory process generates ROS, which may further deteriorate the wound and impede wound tissue regeneration [98]. Thus, ROS scavenging is important in the treatment of BK. In 2022, as a new combined strategy to treat BK, Lin et al. used a PBA–DHPMT-containing glycopolymer to simultaneously kill bacteria, heal wounds, and scavenge ROS [69]. They prepared an amphipathic polymer through the random copolymerization of the PBA–DHPMT monomer and 2-lactobionamidoethyl methacrylate (Figure 6, up).

The resulting polymer self-assembled to form well-defined nanoparticles in the aqueous solution; the nanoparticles can conjugate with the bacteria and aggregate them into clusters due to the borate ester bonds formed by PBA groups and diols on the bacterial cell wall. Lin et al. further encapsulated levofloxacin (LEV, an antimicrobial drug) and chondroitin sulfate (CS, an anti-inflammatory drug) into the nanoparticles to realize the synergistic treatment of BK (Figure 6, up). The drug-carried nanoparticles were almost nontoxic to mammalian cells and can be effectively internalized by human corneal epithelial cells, indicating their ability to penetrate the cornea. With *S. aureus* as a model pathogen, the IC50 of the drug-carried nanoparticles was 17.5 μg/mL, which was half the value of the LEV (35 μg/mL). In the following in vivo experiment, the drug-carried nanoparticles showed the best efficacy on the BK model rats compared to LEV and CS. Due to the synergistic effects of the bactericidal activity of LEV, the wound-healing effect of CS, and the antioxidant capacity of DHPMT, the drug-carried nanoparticles can significantly decrease the intensity of inflammatory factors and relieve the extent of lesions in BK (Figure 6, down). This study demonstrates the competency of the Biginelli reaction in developing multi-functional polymeric bio-platforms.

Besides antioxidation, DHPMT derivatives have been studied as potential antitumor molecules due to their mitosis inhibition activities [99,100,101]. For example, Monastrol, which can be facilely prepared from *m*-hydroxy benzaldehyde through the Biginelli reaction, specifically perturbs the Eg5 kinesin required for spindle bipolarity [99]. However, the poor water solubility and instability of organic small molecules in the bio-system lead to their poor bioavailability [102]. To overcome these problems, in 2020, Tao et al. used the Biginelli reaction to develop anticancer polymers based on DHPMT structures [61]. They prepared four different monomers containing DHPMT moieties, and then copolymerized these monomers with PEGMA to obtain water-soluble polymers, namely, P1–P4 (Figure 7a).

The results of molecular dynamics simulations revealed that these polymers may have inhibitory abilities similar to Monastrol. Tao et al. further studied the anticancer mechanism of these polymers at a cellular level (Figure 7b,c). The normal cells (with the L929 cell as the model) and cancer cells (with the SMMC-7721 cell as the model) were parallelly cultured with the optimized polymer P4. P4 exhibited good cytosafety towards the L929 cells (Figure 7b, up), while the mitotic activity was much like that of the blank control group in the culture medium only (Figure 7c, up). On the contrary, for the SMMC-7721 cancer cells, P4 showed an inhibitory effect without killing them (Figure 7b, down). The chromatins of SMMC-7721 did not move to the poles but were evenly radially distributed around the microtube protein, suggesting the specific inhibitory ability of P4 towards the mitosis of cancer cells (Figure 7c, down). This study opens a new avenue for developing anticancer polymers and verifies the great potential of the Biginelli reaction for identifying new biomedical polymers.

As an efficient and robust coupling tool, the Biginelli reaction has also been used to develop biocompatible polymers with aggregation-induced emission (AIE) properties, which can be applied for bioimaging. In 2018, by combining RAFT polymerization and the Biginelli reaction, Zhang et al. prepared new cross-linked AIE nanoparticles for cell imaging (Figure 8a) [57]. They copolymerized PEGMA and AEMA by RAFT polymerization. An AIE-active dye 4′,4″-(1,2-diphenylethene-1,2-diyl)bis([1,10-biphenyl]-4-carbaldehyde) (CHO-TPE-CHO) and urea were then used to cross-link the polymer through the Biginelli reaction. The resulting amphipathic polymer had a relatively low critical micelle concentration (0.016 mg/mL). The polymer self-assembled in the aqueous solution to form spherical nanoparticles with high dispersity and strong green fluorescence at 510 nm (excitation wavelength = 379 nm).

In the following cell experiment, the fluorescent nanoparticles exhibited good biocompatibility and localized in the cytoplasm and nucleus, indicating that the nanoparticles could be potential biological-imaging reagents (Figure 8b). This study demonstrates that the Biginelli reaction is useful for the synthesis of polymeric bio-probes.

For another example, in 2021, Zhou et al. reported a thermal-responsive dual-modal supramolecular probe for fluorescence imaging and magnetic resonance imaging (MRI) via the Biginelli reaction [68]. A tri-block polymer was prepared by the copolymerization of propargyl methacrylate, *N*-isopropylacrylamide (NIPAM), and AEMA; a two-step orthogonal post-polymerization modification (PPM) was conducted through the azide-alkyne click reaction and the Biginelli reaction to introduce *β*-cyclodextrin and triphenylethenylbenzene (TPE) groups in the side chains, respectively. Through the supramolecular interaction between *β*-cyclodextrin and the adamantane group, Zhou et al. then bound a Gd-based ^1^H MRI contrast agent to obtain the final polymeric probe (PP, Figure 9a). The PP was amphipathic and formed aggregates at a low concentration (0.15 mg/mL). The polymer aggregates were thermally responsive due to the thermally sensitive poly(NIPAM) segment.

With AIE properties and good biocompatibility, the PP was successfully used as a fluorescent thermometer for living cell imaging with bright green fluorescence (Figure 9b). In terms of ^1^H MRI, the calculated relaxation rate of the PP (38.7 × 10^−3^ L/(mol·s)) is approximately 6.67 times that of a common Gd-based MRI contrast agent Magnevist (5.8 × 10^−3^ L/(mol·s)), suggesting a lower concentration of Gd^3+^ was required for the PP to achieve the same contrast effect with Magnevist. Moreover, the PP exhibited relatively low biotoxicity and a long circulation time as a polymeric contrast agent. The MRI of the mouse kidneys was realized using the PP as a contrast agent with a comparable signal intensity to Magnevist at an equivalent concentration of Gd^3+^ (0.5 mmol/L) (Figure 9c). This study provides a new method for synthesizing multi-functional, dual-modal imaging probes, which also reflects the ability of the Biginelli reaction to develop multi-functional polymers for biomedical use.

## 3. Other Applications

Besides bioactive polymers, interesting functions stemming from the DHPM(T) group have also been employed in materials science and polymer science.

The Biginelli reaction has been used to develop polymers with special physical properties. The introduction of rigid DHPM groups can effectively improve the glass transition temperature (*T*_g_) of polymers. In 2016, Meier et al. synthesized dialdehyde and di-*β*-keto esters as monomers with bio-based renewable materials, and then conducted polycondensation to prepare four different polymers (poly(DHPM)-1 to poly(DHPM)-4) through the Biginelli reaction (Figure 10a) [22]. The investigation of the *T*_g_s has led to the identification of some polymers that possess a *T*_g_ as high as 203 °C (Figure 10b), suggesting that the Biginelli reaction is a powerful tool for endowing polymers with new functions. Also in 2016, Tao et al. used the Biginelli reaction to develop an HTP polycondensation strategy [23]. A library of 64 polymers was facilely prepared with *T*_g_s ranging from 50 °C to 159 °C. They also drew *T*_g_ maps of these polymers to successfully predict the *T*_g_s of unprepared polymers. Furthermore, in 2020, by tuning the structure of the di-*β*-keto ester and urea moieties, Meier et al. designed a series of poly(DHPM)s via Biginelli polycondensation to study the structure–property relationship of polymers, where the *T*_g_s of poly(DHPM)s exhibited a gradient distribution from 160 °C to 308 °C [62]. Recently in 2021, Meier et al. reported the PPM of a homopolymer with a long aliphatic side chain through the Biginelli reaction [65]. They found that the *T*_g_s of the polymers dramatically increased by up to 80 °C after PPM. These studies demonstrate that the Biginelli reaction is an effective tool for synthesizing polymers with special thermal stability.

The Biginelli reaction is green, effective, and robust; it has been used as a reliable tool for polymer modification. In 2019, Sui et al. reported using the Biginelli reaction to modify cellulose [59]. They used cellulose acetoacetate (CAA) as the substrate and chose various benzaldehyde derivatives to modify cellulose via the Biginelli reaction (Figure 11). The functionalized polymers showed better thermal stability than CAA and good solubility in the selected solvents; they also possessed various functional groups in their side chains. In 2020, through the Biginelli reaction, Meier et al. also realized the PPM of starch acetoacetate with different renewable aldehydes (benzaldehyde, vanillin, and *p*-anisaldehyde) [60]. These studies demonstrate that the Biginelli reaction is an effective tool for endowing natural polymers with new properties and functions.

In addition, the Biginelli reaction can also efficiently modify polyureas (PUs). In 2021, Deng et al. used the Biginelli reaction to modify PU, successfully upgrading a non-AIE-active PU to polymers with AIE properties [67]. They used commercial dibutylamine and *L*-lysine ethyl diisocyanate to obtain the PU, and selected ethyl acetoacetate and various aliphatic aldehydes to modify the PU via the Biginelli reaction (Figure 12a). The modification reaction was efficient and smooth; the DMSO solutions of the resulting polymer PDHPMs showed no fluorescence under natural light (Figure 12b) but were fluorescent under UV = 365 nm (Figure 12c). Their fluorescence intensity was enhanced by 35-fold at 450 nm compared to the PU. Moreover, the fluorescence intensity of the solutions of the daughter polymers increased with the proportion of the poor solvent (H_2_O) in the mixed solvent (DMSO/H_2_O), indicating their AIE properties. This study reveals the ability of the Biginelli reaction to exploit AIE polymers.

Additionally, the Biginelli reaction has been used to modify carbon nanotubes to facilely prepare carbon–polymer composites [55]. Biginelli-type polymeric adhesives have been developed based on the strong interaction between DHPM groups and metal surface [21,56]. These results all demonstrate that the Biginelli reaction is a powerful tool that can be used to prepare materials with intriguing properties in interdisciplinary fields.

## 4. Perspectives and Conclusions

The Biginelli reaction in polymer chemistry still has a lot of space for further exploration. In the future, polymer synthesis via the Biginelli reaction may focus on several of the following directions. (1) Large-scale preparation. Aldehydes, *β*-keto esters, and (thio)ureas are all common raw materials; thus, the Biginelli-type monomers and polymers may be produced on a large scale, which requires an upgradation of the synthesis methods. An aqueous-phase synthesis system and solvent-free synthesis system are worth trying. (2) Combination with different polymerization methods. Besides free radical polymerization, RAFT polymerization, and ultra-fast RAFT polymerization, other methods (atom-transfer radical polymerization (ATRP), ring-opening polymerization (ROP), sulfur-free RAFT polymerization, etc.) may also be compatible with the Biginelli reaction in order to generate polymers with new main-chain and side-chain structures. (3) The exploration of new polymers on the basis of the reactivity of the DHPMT groups in polymers. DHPMT groups have similar reactivity with thiourea; the substitution reactions between DHPMT and halides have been used to develop new functional polymers [20,64]. Other reactions based on the reactive DHPMT groups should be studied and used to develop new polymers.

In terms of functions, DHPM(T) derivatives are potential anti-bacterial, anti-viral, antiparasitic, and anti-inflammatory drugs [53,54,103]; research on Biginelli-type polymers relating to hygiene might be a promising direction alongside the growing attention to public health during the COVID-19 pandemic. Meanwhile, recent studies revealed that theoretical calculations are conducive to revealing the relationship between polymers’ structures and their functions [31,61,104]. Thus, using theoretical calculations during the preparation of new Biginelli-type polymers may reduce the workload of synthesis and guide the development of new functional polymers.

In this mini-review, we summarized the development of biomedical polymers through the Biginelli reaction, including antioxidant polymers, anticancer polymers, bioimaging agents, etc. Moreover, we also described some other applications of the Biginelli reaction used to generate new functional polymers with intriguing properties (such as high *T*_g_s, AIE, etc.). Besides the pioneering synthesis methodology, these studies represent a deep and comprehensive understanding of DHPM(T) structures for developing new polymers, which may prompt a focus on the functions of MCR structures while developing polymers produced via MCRs.

## Figures and Tables

**Figure 1 molecules-27-07886-f001:**
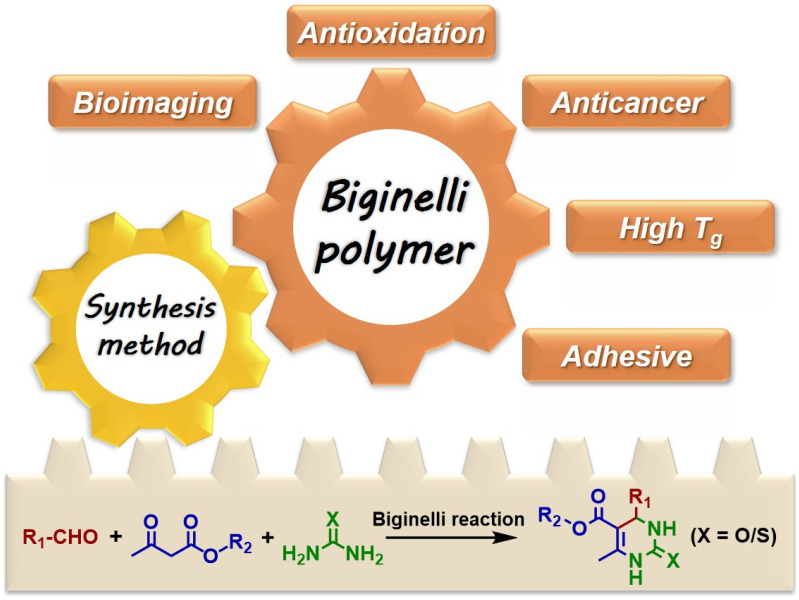
The Biginelli reaction in polymer chemistry.

**Figure 2 molecules-27-07886-f002:**
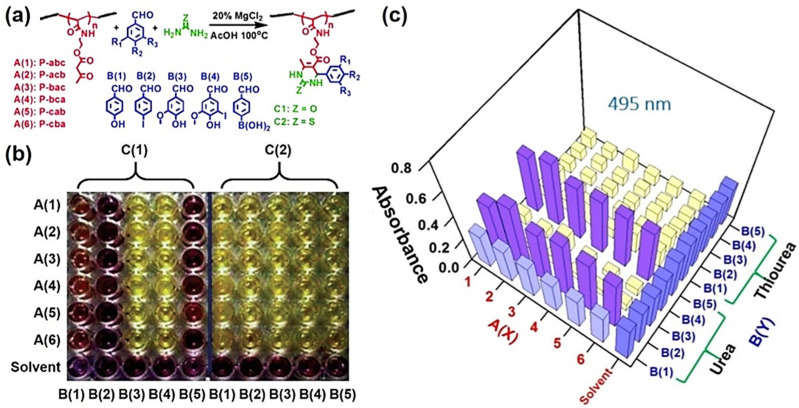
The high-throughput synthesis and screening of Biginelli-type polymers. (**a**) High-throughput synthesis of 60 polymers; (**b**) high-throughput screening of the radical-scavenging ability of the polymers. Polymer solution samples in a 96-well plate (100 µL, 5 mg/mL in CH_3_CN/H_2_O (1/1)) with CH_3_CN/H_2_O (1/1) solvent as the control, with the subsequent addition of the DPPH radical (10 µL, 20 mg/mL in CH_3_CN/H_2_O (1/1)); (**c**) UV absorbance of polymer solutions (100 µL, 5 mg/mL in CH_3_CN/H_2_O (1/1)) at 495 nm after adding DPPH radical. Reprinted with permission from ref. [25]. Copyright © 2017 Royal Society of Chemistry. All rights reserved.

**Figure 3 molecules-27-07886-f003:**
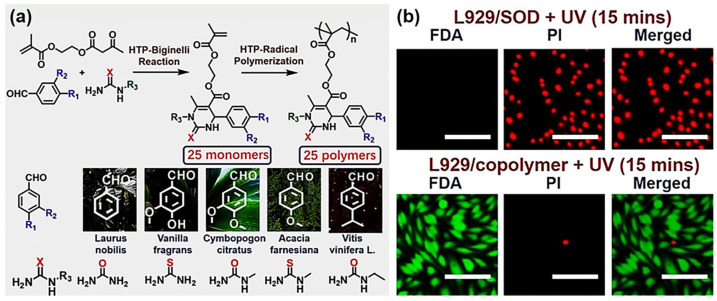
The mini-HTP synthesis of Biginelli polymers and their cell protection along with SOD against UVR: (**a**) HTP methods used to prepare libraries of 25 new monomers and homopolymers; (**b**) fluorescein diacetate/propidium iodide (FDA/PI; green—alive; red—dead) double-staining of L929 cells with UV radiation (15 min) in the presence of SOD (10 mg/mL, up) and copolymer (10 mg/mL, down); scale bar = 100 µm. Reprinted with permission from ref. [26]. Copyright © 2018 American Chemical Society. All rights reserved.

**Figure 4 molecules-27-07886-f004:**
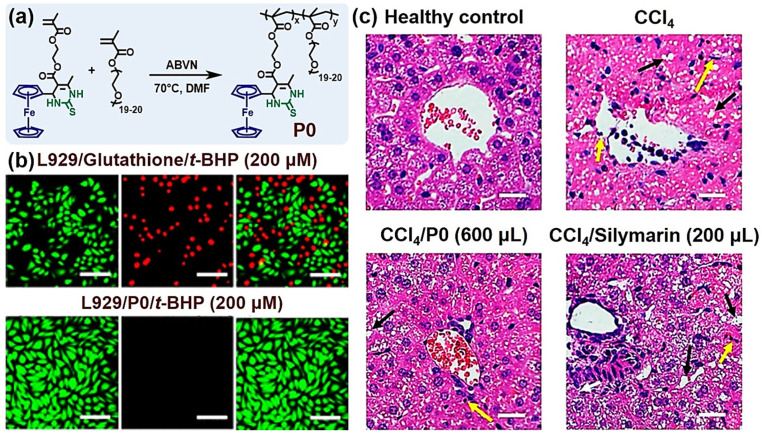
Antioxidant DHPMT–ferrocene polymer (P0) produced via the Biginelli reaction. (**a**) The synthesis of P0; (**b**) FDA/PI double-staining of L929 cells in *t*-BHP + glutathione (200 µM, **up**) and *t*-BHP + P0 (200 µM, **down**); scale bar = 100 µm; (**c**) Histological images of liver sections from healthy control group (**top left**), no additional treatment group (**top right**), CCl_4_/P0 group (600 µL, 20 mg/mL, intravenous tail injection, **bottom left**), and CCl_4_/Silymarin group (200 µL, 20 mg/mL, intraperitoneal injection, **bottom right**); black arrows for vacuoles and yellow arrows for inflamed cells, and scale bar = 20 µm. Reprinted with permission from ref. [58]. Copyright © 2019 American Chemical Society. All rights reserved.

**Figure 5 molecules-27-07886-f005:**
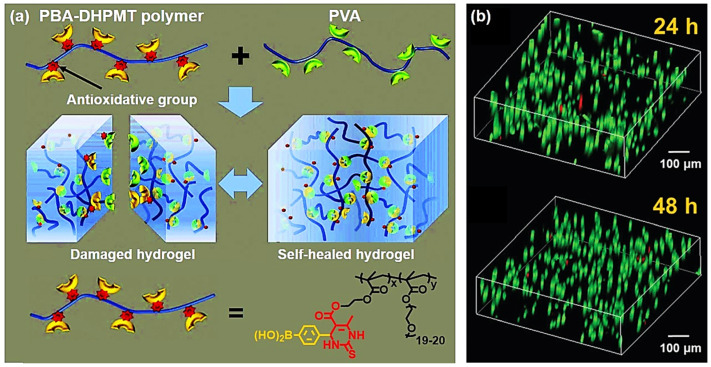
An antioxidant self-healing hydrogel produced via the Biginelli reaction. (**a**) The preparation of the antioxidant hydrogel based on the dynamic borate ester bonds and its self-healing property; (**b**) FDA/PI double-staining of L929 cells in the 3D hydrogel culture matrix (**top**: 24 h; **bottom**: 48 h). Reprinted with permission from ref. [63]. Copyright © 2020 Royal Society of Chemistry. All rights reserved.

**Figure 6 molecules-27-07886-f006:**
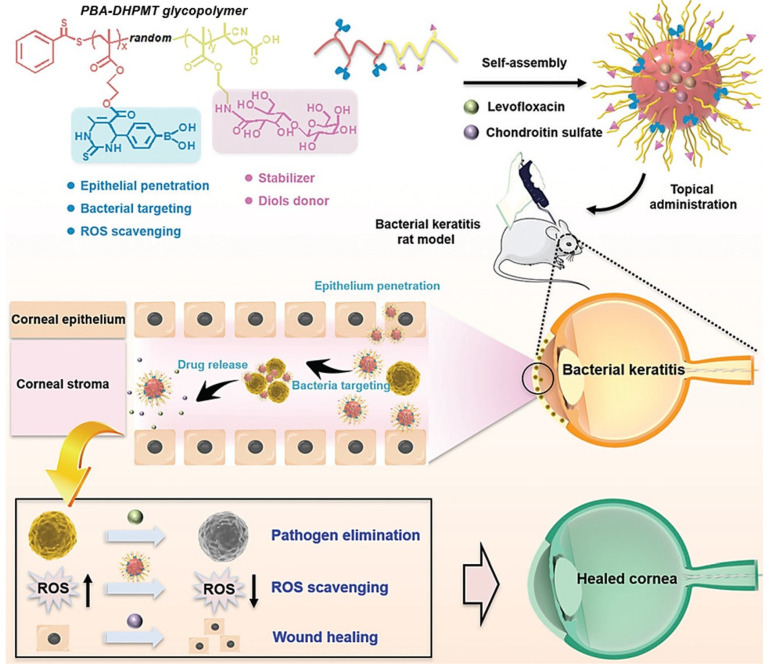
The multi-functional PBA–DHPMT-containing glycopolymeric nanoplatform (**up**) and the therapeutic mechanism of bacterial keratitis in a rat model that combines bactericidal, ROS-scavenging, and wound-healing properties (**down**). Reprinted with permission from ref. [69]. Copyright © 2022 Royal Society of Chemistry. All rights reserved.

**Figure 7 molecules-27-07886-f007:**
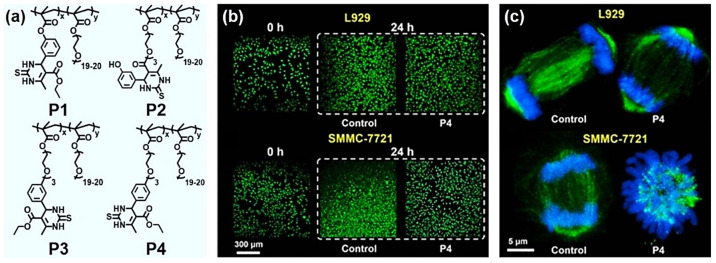
DHPMT-type anticancer polymers produced via the Biginelli reaction. (**a**) The molecular structures of DHPMT-type polymers P1–P4; (**b**) FDA/PI images of L929 (**up**) and SMMC-7721 (**down**) cells incubated with/without P4 (5 mg/mL), 24 h; (**c**) confocal images of L929 (**up**) and SMMC-7721 (**down**) cells incubated with/without P4 (5 mg/mL) using FITC-labeled antibody/Hoechst 33342 for cell staining (green: microtube; blue: chromatin). Reprinted with permission from ref. [61]. Copyright © 2022 American Chemical Society. All rights reserved.

**Figure 8 molecules-27-07886-f008:**
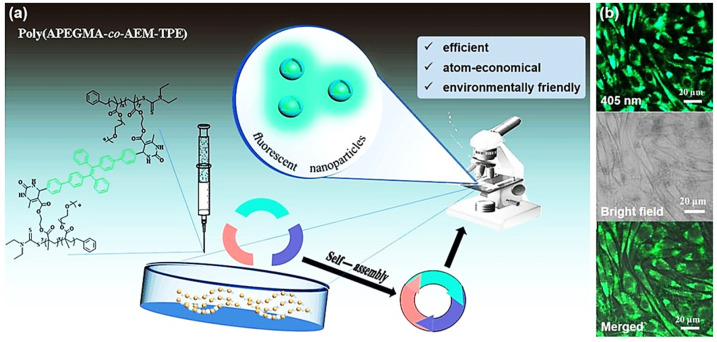
Cross-linked-aggregation-induced emission (AIE) probe for cell imaging via the Biginelli reaction. (**a**) The molecular structure and properties of the AIE probe poly(PEGMA-co-AEMA-TPE); (**b**) confocal images of L929 cells incubated with poly(PEGMA-co-AEMA-TPE) (30 µg/mL) for 3 h (**top**: the fluorescent image of L929 cells under a 405 nm laser; **middle**: the bright field image; **bottom**: the merged image). Reprinted with permission from ref. [57]. Copyright © 2018 Elsevier Inc. All rights reserved.

**Figure 9 molecules-27-07886-f009:**
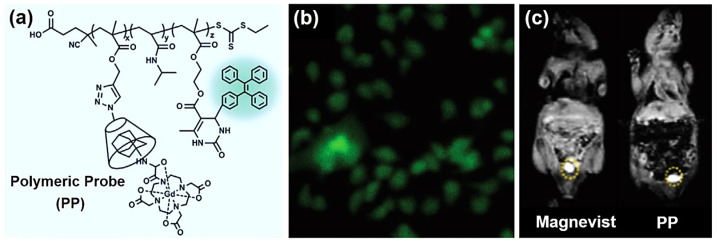
Dual-modal polymeric probe (PP) for fluorescence cell imaging and ^1^H magnetic resonance imaging (MRI). (**a**) The molecular structure of PP; (**b**) fluorescence images of HeLa cells stained with PP solution (50 µg/mL). (**c**) The *T*_1_-weight MRI of mouse kidneys with Magnevist and PP (150 µL, equivalent 0.5 mmol/L Gd^3+^, and intravenous injection) as contrast agents. Reprinted with permission from ref. [68]. Copyright © 2021 Wiley-VCH GmbH. All rights reserved.

**Figure 10 molecules-27-07886-f010:**
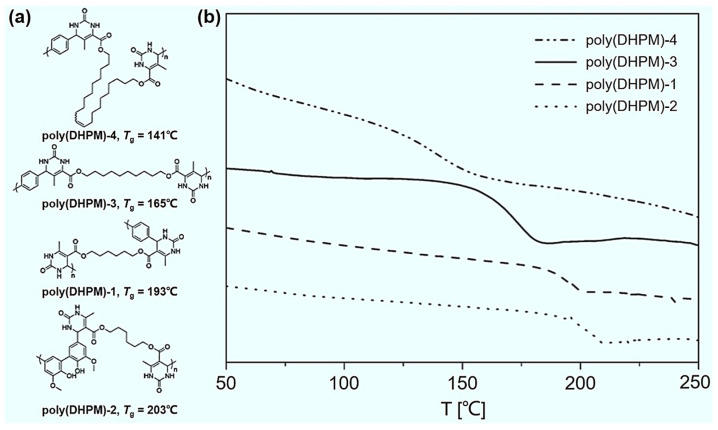
High-glass transition temperature (*T*_g_) renewable polymers produced via the Biginelli polycondensation. (**a**) The molecular structures of four different polymers (poly(DHPM)-1 to poly(DHPM)-4); (**b**) the corresponding differential scanning calorimetry analysis of poly(DHPM)-1 to poly(DHPM)-4. Reprinted with permission from ref. [22]. Copyright © 2016 WILEY-VCH Verlag GmbH & Co. KGaA, Weinheim. All rights reserved.

**Figure 11 molecules-27-07886-f011:**
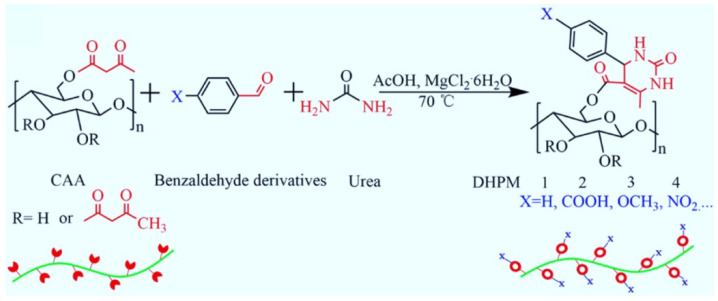
Modification of cellulose acetoacetate (CAA) via the Biginelli reaction. Reprinted with permission from ref. [59]. Copyright © 2019 Elsevier Ltd. All rights reserved.

**Figure 12 molecules-27-07886-f012:**
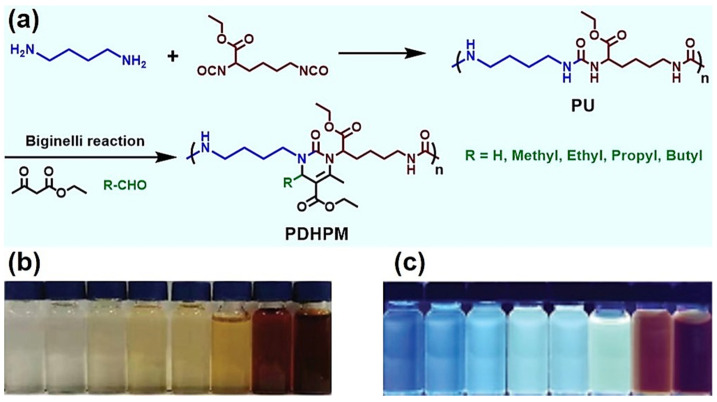
AIE-active polymers (PDHPM) from a non-AIE-active polyurea (PU) produced via the Biginelli reaction. (**a**) The synthesis of PU and PDHPMs; (**b**,**c**) PDHPM (R = propyl) in DMSO solutions with different concentrations (from left to right (mg/mL): 0.05, 0.1, 0.25, 0.5, 1, 5, 10, and 20) under (**b**) natural light and (**c**) UV = 365 nm. Reprinted with permission from ref. [67]. Copyright © 2021 Wiley-VCH GmbH. All rights reserved.

## Data Availability

Not applicable.

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
