# Peer review of "Stepping Further from Coupling Tools: Development of Functional Polymers via the Biginelli Reaction"

_molecules, 2022, doi:10.3390/molecules27227886_

Round 1

Reviewer 1 Report

This review describes development of functional polymers by using Biginelli Reaction.  The review seems to be suitable for publication, but I think some correction should be done.

1. The same naming of the polymer may confuse a reader. I think that the polymers should be named separately one by one.
For example, "P4" in Figure 7, in Figure 9, and in Figure 10 are all different.
I think other symbols of the polymers should be checked.

2. As there may be a mistake because the text is long, I think the sentence should be checked again.

Author Response

1) The same naming of the polymer may confuse a reader. I think that the polymers should be named separately one by one. For example, "P4" in Figure 7, in Figure 9, and in Figure 10 are all different. I think other symbols of the polymers should be checked.

Answer: Many thanks for the professional advice. We have renamed the polymers in the revised article as suggested. Many thanks!

2) As there may be a mistake because the text is long, I think the sentence should be checked again.

Answer: Thanks for the kind suggestion. We have carefully checked the manuscript to correct some typos and grammar mistakes.

Reviewer 2 Report

In the manuscript (molecules-2024283), Ma and coworkers gave a detailed introduction to the applications of the Biginelli reaction in polymer chemistry to identify functional polymers. This review is important for readers who are interested in new applications of multicomponent reactions (MCRs) and functional polymers developed through MCRs. The article is well-documented. Besides the summary of research articles, the authors also gave their perspectives on the future development of the Biginelli reaction in polymer chemistry, which may improve the deep study of this reaction and other MCRs in interdisciplinary fields. I'd like to suggest a minor revision before acceptance for publication. The detailed comments are listed as follows:

1. The mechanism of the Biginelli reaction should be added for a better understanding of readers.

2. The authors should increase the legibility of some figures. For example, the red words in figure 8a are not clear; the proportions of figures 12a and 12b are incongruous.

3. The page number of ref. 15 was missed.

Author Response

1) The mechanism of the Biginelli reaction should be added for a better understanding of readers.

Answer: Many thanks for the suggestion. We have added some descriptions of the mechanism of the Biginelli reaction in the revised manuscript as required.

2) The authors should increase the legibility of some figures. For example, the red words in figure 8a are not clear; the proportions of figures 12a and 12b are incongruous.

Answer: Thanks for the profession suggestion. Figure 8a and Figure 12 have been adjusted for better legibility in the revised manuscript. Thank you!

3) The page number of ref. 15 was missed.

Answer: Thanks for your careful reading! The page number of ref. 15 has been added in the revised manuscript. Many thanks!